

# CFSAN SNP Pipeline 2 (CSP2): a pipeline for fast and accurate SNP distance estimation from bacterial genome assemblies

Robert Literman[1], Jayanthi Gangiredla[2], Hugh Rand[1] and James B. Pettengill[1]

[1] Human Foods Program—Office of Surveillance Strategy & Risk Prioritization—Division of Surveillance and Data Integration, United States Food and Drug Administration, College Park, Maryland, United States
[2] Human Foods Program—Office of Laboratory Operations & Applied Science—Division of Food Safety Genomics, United States Food and Drug Administration, Laurel, Maryland, United States

## ABSTRACT

**Background:** Accurate genetic distance estimation from pathogen whole-genome sequence data is critical for public health surveillance, and with respect to food safety it provides crucial information within traceback and outbreak investigations. The computational demands required for contemporary bioinformatics pipelines to extract high resolution single nucleotide polymorphisms (SNPs) grow in parallel with the size of pathogen clusters, where single strains of common pathogens such as *Escherichia coli* and *Salmonella enterica* can now contain hundreds or thousands of isolates.

**Methods:** To facilitate rapid analysis of whole-genome sequencing (WGS) data for large clusters of foodborne bacterial pathogens, we introduce the CFSAN SNP Pipeline 2 (CSP2). CSP2 is a bioinformatics pipeline coded in Nextflow and Python that extracts SNPs directly from genome assemblies in seconds through rapid MUMmer whole-genome alignment and parallel processing. After genome alignment, most data processing steps mirror the quality control measures used in the CFSAN SNP Pipeline (CSP1), including density filtering and missing data handling.

**Results:** Analysis of simulated data finds that high quality assemblies from the strategic K-mer extension for scrupulous assemblies (SKESA) contain sufficient information for accurate, high resolution SNP distance estimation, while assemblies from the St. Petersburg genome assembler (SPAdes) contained more false positives. CSP2 SNP distances for 150 real-world clusters (50 each of *E. coli*, *Listeria monocytogenes*, and *S. enterica*) were highly correlated with those from CSP1 and the National Center for Biotechnology Information (NCBI) Pathogen Detection pipeline (*E. coli* r >= 0.98; *Salmonella* r = 0.99, *Listeria* r = 0.99). This evaluation of CSP2 demonstrates its comparability to accepted methods and validates its use within future traceback and outbreak investigations.

Corresponding author
Robert Literman,
Robert.Literman@fda.hhs.gov

## INTRODUCTION

High resolution analysis of whole-genome sequencing (WGS) data is an indispensable tool supporting the global surveillance of pathogens. Within food safety, WGS analysis is an integral component in traceback and outbreak investigations where genetic similarity among human clinical cases, environmental samples, and samples from foods and food production environments are inferred to identify potential sources of foodborne illness (*Pightling et al., 2018*; *Franz, Gras & Dallman, 2016*). Many genetic distance estimation pipelines detect and filter single nucleotide polymorphisms (SNPs) after mapping WGS reads against a reference genome, including the widely used CFSAN SNP Pipeline (CSP1) (*Davis et al., 2015*). CSP1 has been effective at uncovering the sources and modes of transmission for many pathogenic outbreaks (*e.g.*, *Whitney et al., 2021*; *Pereira et al., 2023*), but increasing global sequencing efforts over the 10 years since CSP1 was created have resulted in pathogen clusters that now routinely contain thousands of closely related isolates. CSP1 analyses of these 'new normal' larger clusters can be time-consuming if not computationally prohibitive, and the intermediate mapping files (*e.g.*, SAM, BAM, and pileup files) can take up hundreds of gigabytes for moderately sized datasets. As pathogen cluster sizes continue to grow, the ultimate impact of these issues will scale in parallel.

Short-read genome assemblers like the St. Petersburg genome assembler (SPAdes) (*Bankevich et al., 2012*) and strategic K-mer extension for scrupulous assemblies (SKESA) (*Souvorov, Agarwala & Lipman, 2018*) can produce high-quality bacterial genome assemblies in minutes. The high efficiency and accuracy of those assemblers means that genome assembly has become a routine part of pathogen data quality control (*e.g.*, through assessment of assembly length, contig count, completeness, *etc.*), often performed automatically after read data is generated. Their small file size relative to the size of raw reads makes assemblies a computationally efficient representation of genomes from which to infer genetic relatedness. Pipelines like BactSNP (*Yoshimura et al., 2019*) and snippy (https://github.com/tseemann/snippy) already make use of assembly data to infer SNPs; however, both accomplish this by simulating read data from the assemblies. Alternatively, rapid genome aligners like MUMmer (*Delcher, Salzberg & Phillippy, 2003*) can directly align contigs rapidly and accurately in seconds, and the haploid nature of most bacterial species means that high quality genome assemblies will contain sufficient data to infer accurate SNP distances without the need for read data or read simulation.

Here, we introduce CFSAN SNP Pipeline 2 (CSP2), a rapid, high resolution SNP distance estimation pipeline that requires only genome assembly data. Analyses using CSP2 finish in a fraction of time required for CSP1, due in large part to efficient parallelization and the application of MUMmer to quickly and accurately align assemblies. To explore the impact of assembler choice and assembly read depth on CSP2 SNP detection, we evaluate the recovery of SNPs simulated at known genomic locations from SPAdes and SKESA assemblies. To ensure concordance with results from CSP1 (*Davis et al., 2015*) and the National Center for Biotechnology Information (NCBI) Pathogen Detection pipeline (*NCBI Insights, 2025*; *Sayers et al., 2024*), two pipelines used routinely in support of public health initiatives around the world, we compare distance estimates

involving over 11,000 isolates deriving from 50 clusters each of *Escherichia coli*, *Salmonella enterica*, and *Listeria monocytogenes*.

## MATERIALS AND METHODS

### CSP2 installation

CSP2 is an open-source pipeline and available to download at www.github.com/CFSAN-Biostatistics/CSP2. CSP2 is written in a combination of Nextflow and Python and has the following software dependencies that can be installed manually or handled automatically *via* a supplied Conda recipe: Nextflow (v23.04.3) (*Di Tommaso et al., 2017*), Python (v3.8.1), MUMmer (v4.0.0) (*Delcher, Salzberg & Phillippy, 2003*), BEDTools (v2.26.0) (*Quinlan & Hall, 2010*), BBTools (v38.94) (www.jgi.doe.gov/data-and-tools/software-tools/bbtools), Mash (v2.1.1) (*Ondov et al., 2016*), and SKESA (v2.5.0) (*Souvorov, Agarwala & Lipman, 2018*). All run parameters (Table S1) can be set on the command line or in the Nextflow configuration files. Nextflow natively manages job submissions on local machines or on computing clusters running SLURM, PBS/Torque, or Amazon Web Services (AWS) Batch.

### CSP2 architecture

The general workflow of CSP2 is shown in Fig. 1. CSP2 comparisons among isolates are currently performed entirely at the assembly level; if reads are provided, a SKESA assembly is generated which is used for downstream analysis. CSP2 aligns each query genome assembly to one or more reference genomes using the *dnadiff* function from MUMmer. For each alignment, CSP2 compiles multiple MUMmer outputs into a single *snpdiffs* file containing three main sections: (1) a header line with assembly and alignment metadata (*e.g.*, sample names, file paths, assembly sizes, contig counts, alignment coverage, kmer similarity), (2) a browser extensible data (BED) section, including all aligned and unaligned regions, and (3) data for each SNP detected by MUMmer, including genomic coordinates, reference and query bases, metrics for the corresponding alignment (*e.g.*, alignment length, percent identity), and a category designation of "SNP" (both reference and query have an ACTG base), "Indel" (reference or query has a gap), or "Invalid" (reference or query has a non ACTG base like 'N' or '?'). Taken together, *snpdiffs* files represent a human- and computer-readable version of each alignment, and like other intermediate mapping files (*e.g.*, binary alignment maps (BAMs), pileups) *snpdiffs* can be reused for any downstream CSP2 analysis eliminating the need for realignment. However, unlike larger mapping-based files *snpdiffs* are typically around 1 Mb.

To ensure consistency with distances inferred by CSP1, the default quality control (QC) and SNP filtering criteria for CSP2 largely mirror those in CSP1. Full CSP2 distance analysis is limited to query genomes that cover a specified percentage of the reference genome (–*min_cov*; default: 85%), and MUMmer SNPs are not considered in final distances if: (1) they include a non-ACTG base or an indel, (2) they occur on a short alignment (–*min_len*; default: 500 bp), (3) they occur on an alignment with low percent identity (–*min_iden*; default: 99%), (4) they occur within a set of window sizes

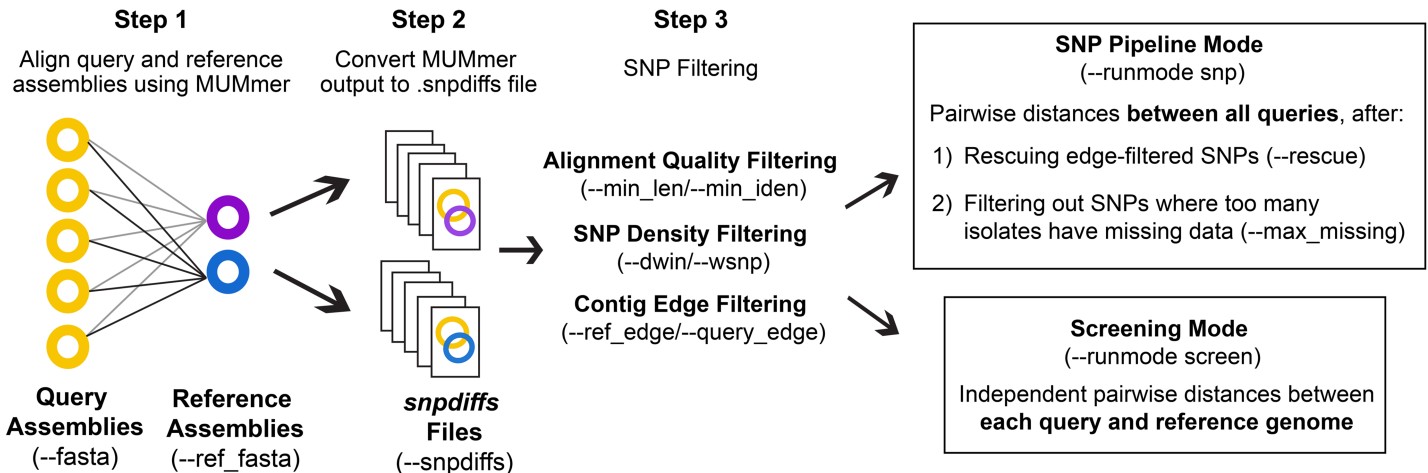

**Figure 1 CSP2 workflow.** CSP2 calls MUMmer to align query and reference assemblies and consolidates alignment results into a lightweight *snpdiffs* file that can be reused in future CSP2 analyses. CSP2 masks SNPs that occur on short or poorly aligned contigs, SNPs in dense clusters, or SNPs near contig edges. In 'Screening' mode, CSP2 reports these filtered SNP distances between each query and reference genome (*e.g.*, for rapid screening of incoming sequencing data against known laboratory control strains). If running in 'SNP Pipeline' mode, CSP2 outputs SNP alignments and pairwise distances between queries based on one or more reference genomes.

(–*dwin*; default: 1000,125,15 bases; disabled by setting to 0) where there are more than –*wsnps* SNPs (default: 3,2,1), or (5) they are the result of multiple alignments to the same locus. All SNPs that pass these conditions are filtered out if they fall within –*ref_edge* or –*query_edge* bases from the edge of a reference or query contig, respectively (default: 350 bp; disabled by setting to 0). If the –*rescue* flag is passed, any SNPs purged due solely to contig edge proximity will be 'rescued' if the same reference position is covered more centrally by another query (*i.e.*, for SNPs that pass all other filters, edge filtering is overruled if there is more internal contig support from another query).

CSP2 can analyze *snpdiffs* files in two run modes: Screening mode (–*runmode* screen) or SNP pipeline mode (–*runmode* snp). In screen mode, query-reference comparisons are analyzed independently and the filtering steps described above mark the end of the run. CSP2 outputs pairwise distances for each query-reference comparison along with general alignment information. In SNP mode, CSP2 considers queries mapped to the same reference to be part of a related group and outputs pairwise distances between query isolates along with SNP alignments in FASTA format. Reference genomes for SNP analyses can be manually specified or chosen automatically; CSP2 selects references based on assembly metrics and mean kmer distances calculated *via* Mash. When multiple references are requested (–*n_ref*; default: 1), CSP2 adds weight to references that capture more of the clade diversity, which helps avoid choosing highly similar references that offer mostly redundant information. To reduce the impact of low-quality datasets that result in spurious alignments and false-positive SNPs, CSP2 can subset distance results by masking SNPs where too many query isolates either have no alignment coverage (*i.e.*, missing data) or have data that was purged during QC (–*max_missing*; default: 50%).

## Simulated data analysis

To explore the impact of assembler choice and assembly read depth on CSP2 performance, we employed the same simulated data used for validation of CSP1 (*Davis et al., 2015*). Briefly, the CFSAN SNP mutator (*Davis et al., 2015*) was used to generate 100 mutated versions of a closed *Salmonella* Agona genome (NCBI RefSeq NC 011149.1); each mutated genome received 300 SNPs, 20 insertions, and 20 deletions at known positions. For each mutated genome, 100X depth Illumina-style reads were simulated using ART (*Huang et al., 2011*) and we subsampled each dataset down to 80X, 60X, 40X, and 20X coverage using *reformat.sh* from BBTools. For each read depth, we reassembled the genomes using SKESA v.2.5.0 and SPAdes v3.15.3 with default options.

Using the screen mode of CSP2, we queried the distances from the unmutated reference genome to each of the 100 mutated genomes, along with the SPAdes and SKESA genome short-read assemblies. To test the utility of rescuing SNPs lost solely due to edge filtering we reanalyzed the resulting *snpdiffs* files using SNP pipeline mode with edge-rescuing enabled. We ran one SNP analysis per mutated genome, where the queries consisted of the 20X–100X assemblies for each mutated genome from either SKESA or SPAdes.

## Real-world data analysis

To compare CSP2 SNP distance estimates to results from CSP1 and the NCBI Pathogen Detection pipeline, we selected 50 clusters each of *E. coli*, *L. monocytogenes*, and *S. enterica* at random from the NCBI Pathogen Detection database, provided they contained between 50–150 isolates (Table S2). Pairwise distances between isolates were calculated using CSP1 and CSP2, and NCBI Pathogen Detection SNP distances were retrieved from their database. NCBI distances are calculated using the patristic distance between tips on a maximum compatibility tree (*Cherry, 2017*), while CSP1 identifies SNPs through read mapping against a reference genome (*Davis et al., 2015*). All three distance estimation methods have various QC strategies that factor into the final distance estimates (*Davis et al., 2015*; *Sayers et al., 2024*; *Cherry, 2017*). CSP2 preserved distances (SNPs that passed all filters, went through edge rescuing, and had alignment coverage in 50% or more assemblies) were compared to the unambiguous delta positions from NCBI and the preserved SNPs from CSP1.

All isolates had an associated SKESA assembly; for each cluster, the SKESA assembly with the highest RefChooser score (www.github.com/CFSAN-Biostatistics/refchooser) was used for CSP1 read mapping. CSP2 analyses were run using four reference genomes: the reference genome used for CSP1 read mapping and three additional references chosen automatically by CSP2. Final CSP2 distances were based on the reference genome where queries had the highest median alignment rate. Exceptionally fragmentary assemblies are often indicative of data quality issues (*e.g.*, low sequencing depth, contamination, high sequencing error rate); we identified and partitioned contig count outliers using an interquartile range (IQR) analysis, flagging isolates where the contig count was higher than 1.5*IQR for the species.

For each pairwise comparison, we calculated and compared SNP distances between CSP2, CSP1, and NCBI. For each SNP cluster, we calculated the slope and correlation

between distances across methods, along with the mean and 95% confidence intervals (CI) around the estimate differences (*i.e.*, based on all pairwise comparisons in Cluster X, the average difference between CSP2 and CSP1 was Y SNPs). For each individual isolate we averaged the between-method differences across pairwise comparisons including that isolate (*i.e.*, for all pairwise comparisons involving isolate X, the average difference between CSP2 and CSP1 was Y SNPs) along with 95% confidence intervals. We also calculated the isolate-specific mean deviations, which were then used to compare methods at the species level.

# RESULTS

## Simulated data

Analysis of the 100 mutated genomes using CSP2 screen mode resulted in recovery of all 300 expected SNPs, insertions, and deletions with no false positives (Table S3). For assemblies created from the reads generated off the mutated reference, contig counts for the SPAdes assemblies were consistent across read depths (64–91 contigs; Table S4; Fig. S1) and these results were in line with the 60X–100X SKESA assemblies (65–88 contigs; Table S4; Fig. S1); SKESA assemblies were less contiguous for the 40X subsets (median: 193 contigs; Table S4; Fig. S1) and 20X subsets (median: 1,675 contigs; Table S4; Fig. S1). SPAdes assemblies also had consistent assembly size across read depths (Raw *S. Agona*: 4.80 Mb; SPAdes: 4.75–4.76 Mb; Table S4; Fig. S1). SKESA assemblies were smaller than their SPAdes counterpart, and while the results from the 40X–100X subsets were consistent (4.70–4.73 Mb; Table S4; Fig. S1), the 20X subsets resulted in smaller assemblies (median: 4.57 Mb; Table S4; Fig. S1).

The CSP2 MUMmer alignments of SPAdes assemblies against the reference genome resulted in a median reference genome coverage of 98.9% (Fig. S2; Table S5). For the 60X–100X SKESA assemblies, the median reference genome coverage was 98.4–98.5%, while the 40X and 20X SKESA assemblies covered less (97.9% and 93.6%, respectively; Fig. S2; Table S5). Query assemblies had a median coverage of 99.8% excluding the 20X SKESA assemblies; less of the 20X SKESA query assemblies aligned to the reference genome after filtering out short or low-quality alignments (median: 98.2%; Fig. S2; Table S5). Among the short-read assemblies, CSP2 screening mode detected 290–300 across SPAdes datasets (median: 297; Table S5), and the 40X–100X SKESA assemblies resulted in a similar SNP recovery rate (288–300; median: 295; Table S5). Fewer SNPs were recovered from the 20X SKESA assemblies (274–295; median: 286; Table S5).

Although most SNPs were detected in the screen mode, some of them failed CSP2 filters and were not designated as 'SNPs'. Edge-filtering (masking SNPs within 350 bp of a contig edge) resulted in the largest removal of true positives, and SKESA datasets were most impacted. SPAdes assemblies lost ~one edge-filtered SNP per assembly across depths, where edge filtering removed ~two SNPs per SKESA assembly for the 60X–100X datasets, ~eight SNPs per assembly for the 40X SKESA datasets, and ~60 SNPs per assembly for the 20X SKESA datasets (Table S6). Beyond missing data and edge filtering, CSP2 purging of valid SNPs was generally rare. The 20X SKESA assemblies lost ~five SNPs per assembly due to other filters (*e.g.*, alignment length, percent identity, density filtering; Table S6), but

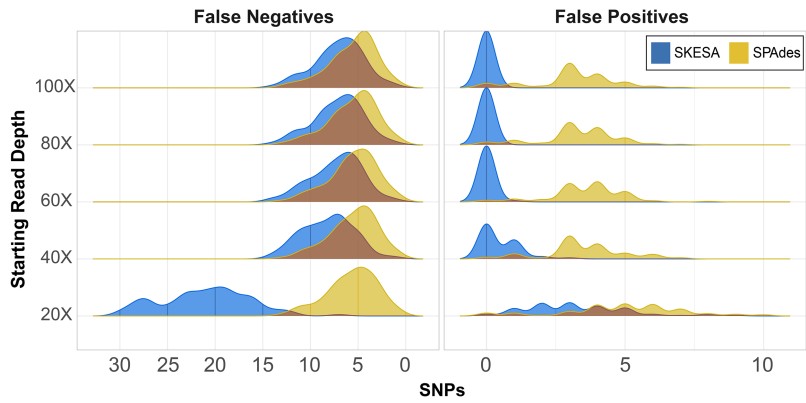

**Figure 2 False positive and negative SNP counts from CSP2 analyses of simulated *Salmonella* genomes assembled using SKESA and SPAdes.** For all starting read depths, CSP2 analysis of SPAdes assembly data (yellow) resulted in consistently low false negative rates (median: five false negatives per assembly). SNP recovery rates were similar for the 40X–100X SKESA assemblies (blue; median: seven false negatives per assembly), but 20X SKESA assemblies were significantly less complete (median: 21 false negatives per assembly). SPAdes assemblies contained a median of four false positive SNPs per assembly, while the median SKESA assembly had no false positives.

for all other depths and both assemblers the median number of true positive SNPs purged from any assembly was zero, and no other SKESA or SPAdes assembly had more than three SNPs that were present and incorrectly filtered out (Table S6).

CSP2 analysis of SPAdes assemblies detected an average of 29–37 false positives per assembly across read depths (Fig. 2; Table S6). CSP2 filters correctly masked between 86–90% of these sites, but false positives called as SNPs averaged three—five per SPAdes assembly across depths, totaling 1,847 (Fig. 2; Table S6). In contrast, SKESA assemblies contained 371 total false positives called as SNPs, and over 86% came solely from the 20X datasets. There were 479 false positives detected across the 20X SKESA assemblies with 322 called as SNPs, and false positive rates dropped significantly at higher depths (40X: 217 detected, 47 called as SNPs; 60X: three detected, two called as SNPs) with no false positives detected in the 80X or 100X SKESA datasets (Fig. 2; Table S6).

In the most fragmentary assembly (SKESA 20X depth), 98% of the true positive SNPs that were edge-filtered during screening mode were restored when analyzed alongside more contiguous assemblies (5,956 rescued; Table S6). For the 40X–80X SKESA datasets, 12–78% of edge-filtered true positive SNPs were restored. Among those SKESA sites that were edge-filtered, some were correctly removed; there were 106 false positives present only in the 20X SKESA assemblies and one among all 60X assemblies, and these sites were correctly not rescued as they occurred in only one of the query genomes. None of the rescued SKESA SNPs were false positives (Table S6), suggesting utility for combining reasonable edge filters and edge rescuing based on context from additional assemblies. Longer SPAdes contigs resulted in less overall edge filtration; 53 total SNPs were rescued across all assemblies (Table S6). However, 42 of the 53 edge-rescued SPAdes SNPs were

**Table 1 Cluster- and isolate-level concordance between SNP distances estimated by CSP2, CSP2, and NCBI Pathogen Detection.** For each species, the lowest average SNP differences came from the CSP2/CSP1 comparison, followed by CSP2/NCBI, then CSP1/NCBI. Mean slopes and correlation values were high across comparisons, and distances for *E. coli* were consistently higher at NCBI.

| Comparison | Species | Mean value across clusters (range of cluster values) | | | Mean SNP difference across isolates [95% CI] | |
| --- | --- | --- | --- | --- | --- | --- |
| | | Slope | Correlation | SNP difference | Contig inlier | Contig outlier |
| CSP1 *vs.* CSP2 | *E. coli* | 1.00 (0.87–1.22) | 0.99 (0.97–1.0) | −0.12 (−2.58–2.53) | −0.18 [−3.25 to 2.89] | 2.34 [−4.57 to 9.26] |
| | *Listeria* | 1.00 (0.93–1.06) | 0.99 (0.96–1.0) | −0.28 (−3.16–1.20) | −0.35 [−2.80 to 2.10] | 0.12 [−3.57 to 3.81] |
| | *Salmonella* | 1.00 (0.91–1.28) | 0.99 (0.93–1.0) | −0.07 (−1.50–1.32) | −0.09 [−2.32 to 2.15] | 0.38 [−4.61 to 5.37] |
| NCBI *vs.* CSP2 | *E. coli* | 1.02 (0.80–1.36) | 0.98 (0.95–1.0) | 1.87 (−3.64–5.59) | 1.80 [−3.19 to 6.78] | 8.45 [−4.43 to 21.3] |
| | *Listeria* | 1.00 (0.93–1.12) | 0.99 (0.96–1.0) | 0.49 (−2.23–4.04) | 0.40 [−2.22 to 3.01] | 3.64 [−3.97 to 11.2] |
| | *Salmonella* | 1.00 (0.72–1.06) | 0.99 (0.88–1.0) | 0.44 (−2.30–1.78) | 0.44 [−2.19 to 3.08] | 2.59 [−3.81 to 9.0] |
| NCBI *vs.* CSP1 | *E. coli* | 1.01 (0.71–1.13) | 0.98 (0.91–1.0) | 1.99 (−3.87–6.64) | 1.98 [−3.22 to 7.17] | 6.56 [−4.78 to 17.9] |
| | *Listeria* | 1.00 (0.90–1.11) | 0.99 (0.93–1.0) | 0.77 (−0.62–3.94) | 0.75 [−2.22 to 3.71] | 3.52 [−4.40 to 11.5] |
| | *Salmonella* | 0.99 (0.42–1.05) | 0.98 (0.70–1.0) | 0.52 (−1.68–2.52) | 0.53 [−2.64 to 3.69] | 2.21 [−3.53 to 7.96] |

false positives suggesting that while were correctly filtered out in some assemblies, they also existed more centrally others (*i.e.*, assembler artefacts).

CSP2 SNP pipeline mode called SNPs at 287–299 positions per SPAdes assembly (median: 295; Table S6), and results were similar for the 40X–100X SKESA datasets (286–299 SNPs; median: 293; Table S6). Fewer SNPs were detected in the 20X SKESA assemblies due to lack of coverage (median: 279; Fig. 2; Table S6). SPAdes assemblies had a median of four false positives per assembly (Fig. 2; Table S6), and around 20% of the CSP2 distance estimates for SPAdes assemblies went beyond the expected 300 SNPs (max: 308 SNPs; Table S6). The median SKESA assembly had no false positives, and no SKESA distances were estimated beyond 300 SNPs (Fig. 2; Table S6).

### Real world data

Contig counts and N50 values for all SKESA assemblies were analyzed at the species level; median contig counts were 51 for *Listeria* (outlier: 194), 119 for *Salmonella* (outlier: 376), and 394 for *E. coli* (outlier: 880) (Table S7; Fig. S3). Contig count outliers were partitioned from the larger dataset prior to downstream analysis including 248 *E. coli*, 452 *Listeria*, and 336 *Salmonella* assemblies. After removing contig count outliers, isolate counts were 3,770 (*E. coli*), 3,458 (*Listeria*), and 3,930 (*Salmonella*), and median N50 values for the remaining assemblies were 237 Kb (*Listeria*), 110 Kb (*Salmonella*), and 77 Kb (*E. coli*) (Table S7).

Median CSP1 read mapping depths were 56 (*E. coli*), 64 (*Salmonella*), and 82 (*Listeria*), and median read mapping rates were 94.4% (*E. coli*), 97.1% (*Salmonella*), and 97.6% (*Listeria*) (Table S7). Median CSP2 reference genome coverages were 97.9% (*E. coli*), 99.3% (*Listeria*), and 99.2% (*Salmonella*) and query coverages were similar with median rates of 98.1% (*E. coli*), 98.6% (*Listeria*), and 99.3% (*Salmonella*) (Table S7). By default, CSP2 does not complete the SNP pipeline analysis for any queries that fail to cover 85% of their reference genome, and there were 174 total alignments from 40 *E. coli*, seven *Listeria*, and five *Salmonella* where this applied.

Cluster-level correlations and slopes between pairwise distances were consistently high across methods and species, illustrating that there is not only a strong linear relationship between the pairwise distances but also that the magnitude of the distance between any two isolates was similar among the methods (Table 1; Table S8). One *Listeria* cluster contained isolates that varied at five total SNP positions (Lm_PDS000060146.5); the average SNP difference between any method was less than 0.25 SNPs, but this cluster was not included in slope or correlation comparisons due to artifacts associated with limited variation. For all three species the average correlation across clusters was either 0.98 or 0.99 for all method comparisons (Table 1; Table S8). Slopes were similarly consistent between species with average slopes across clusters ranging between 0.99–1.02 for any method comparison (Table 1; Table S8).

We calculated the mean SNP difference between methods across all pairwise comparisons for each cluster, and for each species the smallest average difference came from the CSP2/CSP1 comparison, followed by CSP2/NCBI, then CSP1/NCBI (Fig. 3; Table 1). *Listeria* and *Salmonella* clusters were most concordant across methods; the average between-method deviation across *Listeria* clusters was 0.28 SNPs for CSP2/CSP1 (individual cluster average range: −3.1–1.2 SNPs; Fig. 3; Table 1; Table S8), 0.49 SNPs for CSP2/NCBI (cluster range: −2.2–4.0 SNPs), and 0.77 SNPs for CSP1/NCBI (cluster range: −0.6–3.9 SNPs) (Fig. 3; Table 1; Table S8). The average between-method deviation across *Salmonella* clusters was 0.07 SNPs for CSP2/CSP1 (cluster range: −1.5–1.3 SNPs; Fig. 3; Table 1; Table S8), 0.44 SNPs for CSP2/NCBI (cluster range: −2.3–1.8 SNPs), and 0.52 SNPs for CSP1/NCBI (cluster range: −1.7–2.5 SNPs) (Fig. 3; Table 1; Table S8). The CSP2/CSP1 concordance for *E. coli* clusters was in line with results from *Listeria* and *Salmonella* (mean: 0.12 SNPs, cluster range: −2.6–2.5 SNPs; Fig. 3; Table 1; Table S8), but *E. coli* distances from NCBI were generally higher than those from CSP2 or CSP1; distances for the average *E. coli* cluster were 1.9 SNPs higher than CSP2 (cluster range: −3.6–5.6 SNPs) and 2.0 SNPs higher that CSP1 (range: −3.9–6.6 SNPs) (Fig. 3; Table 1; Table S8).

Individual isolates that result in consistently more or less SNPs between methods will yield many discordant pairwise distances; therefore, results were also analyzed by averaging the between-method differences for all pairwise comparisons involving each isolate. For *Listeria* and *Salmonella*, all three method comparisons resulted in an average isolate-level SNP difference less than 1 SNP (Fig. 4; Table S9). The average *Listeria* isolate had a mean difference of 0.35 SNPs between CSP2 and CSP1 (95% CI [−2.8–2.1] SNPs), 0.40 SNPs between CSP2 and NCBI (95% CI [−2.2 to 3.0] SNPs), and 0.75 SNPs for CSP1/NCBI (95% CI [−2.2 to 3.7] SNPs) (Fig. 4; Table 1; Table S9). The average *Salmonella* isolate had a mean difference of 0.02 SNPs for CSP2/CSP1 (95% CI [−2.3 to 2.2] SNPs), 0.31 SNPs for CSP2/NCBI (95% CI [−2.2 to 3.1] SNPs), and 0.53 SNPs for NCBI/CSP1 (95% CI [−2.6 to 3.7] SNPs) (Fig. 4; Table 1; Table S9). As expected based on the cluster-level analysis, NCBI distances for *E. coli* isolates were consistently higher than those from CSP2 and CSP1, and compared to *Listeria* and *Salmonella* the 95% confidence intervals around *E. coli* averages were broader for all comparisons (Fig. 4; Table 1; Table S9). Relative to CSP2 and CSP1, NCBI *E. coli* distances were an average of 1.8 SNPs (95% CI [−3.2 to 6.8] SNPs) and 2.0 SNPs (95% CI [−3.2 to 7.2] SNPs) higher, respectively

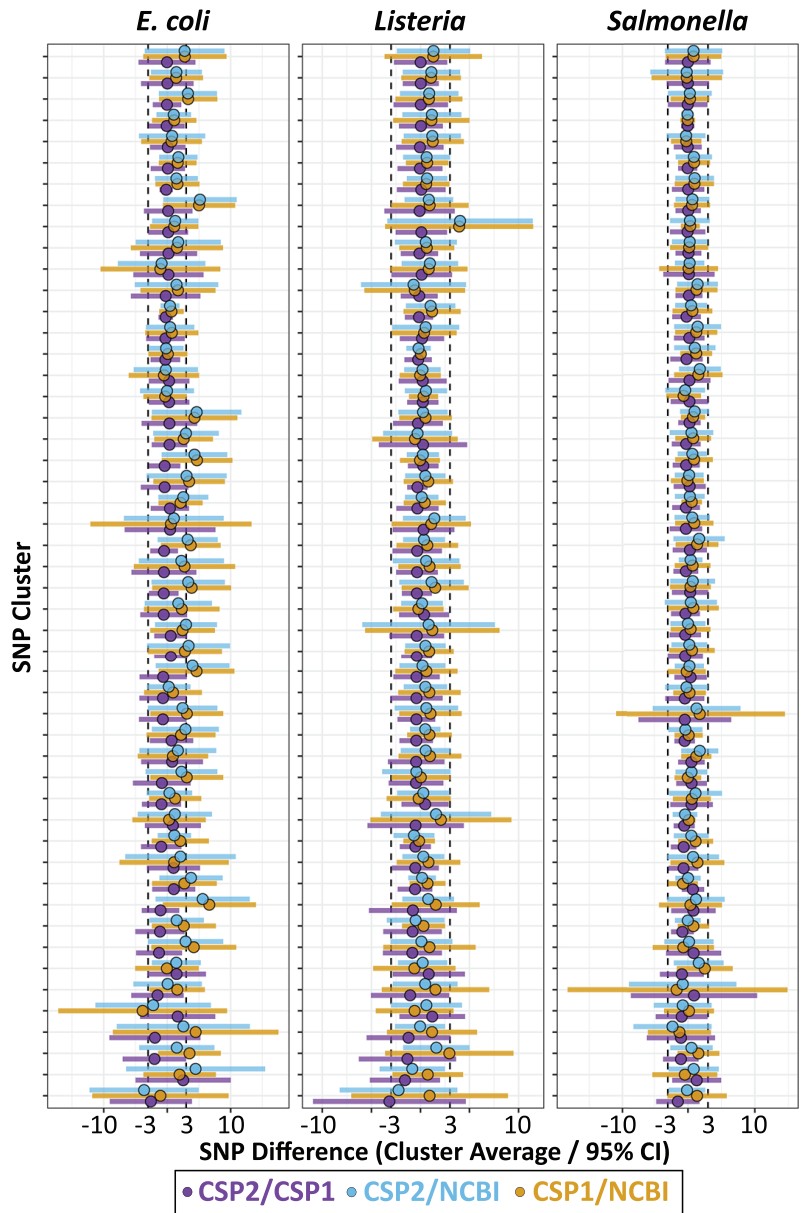

**Figure 3** **SNP distance differences between CSP2, CSP1, and NCBI averaged by cluster.** Across 50 clusters per species (Y-axis), average cluster-level SNP differences between methods (points) and 95% confidence intervals (bars) were smallest when comparing results from CSP2 and CSP1 (purple; mean *E. coli* cluster difference: 0.12 SNPs, *Listeria*: 0.28 SNPs, *Salmonella*: 0.07 SNPs). The average difference between NCBI distances and those from CSP2 (blue) or CSP1 (yellow) were within 0.8 SNPs for *Listeria* and *Salmonella* clusters, but were two SNPs higher on average for *E. coli* clusters (yellow/blue points clustering right of center).

(Table S9); for CSP2/CSP1, the mean isolate-level difference was 0.18 SNPs (95% CI [−3.2 to 2.9] SNPs) (Fig. 4; Table 1; Table S9).

Isolates across species that were classified as contig count outliers resulted in broader confidence intervals for all comparisons, and SNP distance comparisons with NCBI were most affected (Fig. 4; Table S9). For *Listeria* and *Salmonella*, the average CSP2/CSP1

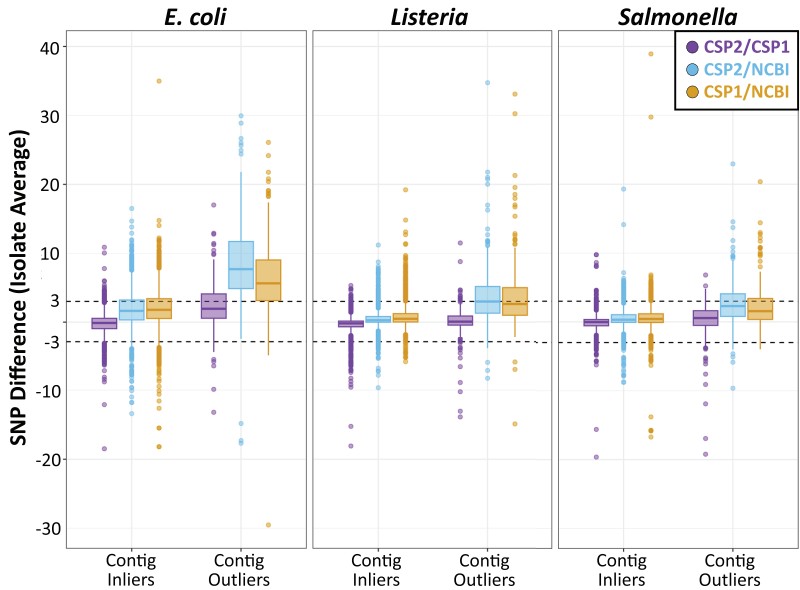

**Figure 4 SNP distance differences between CSP2, CSP1, and NCBI averaged by isolate for more and less contiguous assemblies.** Across species, distances for the average isolate varied least between CSP2 and CSP1 (purple; mean *E. coli*: 0.18 SNPs; mean *Listeria*: 0.35 SNPs; mean *Salmonella*: 0.38 SNPs), followed by CSP2/NCBI (blue), then CSP1/NCBI (yellow). *Salmonella* and *Listeria* isolates had an average difference of 0.75 SNPs or less across methods, but NCBI estimates for *E. coli* isolates were around two SNPs higher than estimates from CSP2 and CSP1. For fragmentary assemblies with more contigs than the threshold outlier value, CSP2 distances were within 0.5 SNPs of CSP1 for *Listeria* and *Salmonella* isolates, but the between-method differences ranged from two—eight SNPs for all other comparisons.

differences for contig count outliers were 0.12 and 0.38 SNPs, respectively (Fig. 4; Table 1; Table S9). While still highly concordant on average, compared to the 95% confidence interval ranges for inliers (*Listeria*: 4.9 SNPs; *Salmonella*: 4.5 SNPs) outlier comparisons were more variable (*Listeria*: 7.4 SNPs; *Salmonella*: 10.0 SNPs; Fig. 4; Table 1; Table S9). *E. coli* contig count outliers resulted in more disparate distance estimates; compared to the CSP2/CSP1 inlier comparisons (mean difference: −0.18 SNPs; CI range: 6.1 SNPs), the average *E. coli* outlier isolate had 2.3 more SNPs in CSP1 compared to CSP2 (95% CI [−4.6–9.3] SNPs; CI range: 13.8 SNPs) (Fig. 4; Table 1; Table S9). For all three species, NCBI distances involving contig count outliers were consistently higher than those from CSP2 or CSP1; *Listeria* contig count outliers had an average of 3.5 and 3.6 more SNPs in NCBI than in CSP2 or CSP1, respectively, and *Salmonella* outliers had an average of 2.2 more SNPs than CSP2 and 2.6 more SNPs than CSP1 (Fig. 4; Table 1; Table S9). The biggest differences were seen in *E. coli*, where on average NCBI outlier isolates contained 8.5 and 6.6 more SNPs than CSP2 and CSP1, respectively (Fig. 4; Table 1; Table S9).

## DISCUSSION

CSP2 generates accurate genetic distance estimates between bacterial pathogens using only genome assembly data. Across over 450,000 pairwise comparisons from *E. coli*, *Listeria*,

and *Salmonella*, the median difference between CSP2 and CSP1 distance estimates was zero. Results were also highly concordant with values from the NCBI Pathogen Detection database, suggesting that CSP2 results are interchangeable with those methods as they apply to pathogen surveillance. When starting from assembled data, the average CSP2 analysis using four reference genomes completed in less time than it took CSP1 to call SNPs for each isolate against one reference reflecting a speed increase that will have real-world implications for outbreak response efforts.

Based on results from the simulated data analysis, it is advised to analyze genome assemblies generated using a single assembly pipeline when possible. Starting from the same simulated read data, SPAdes assemblies contained consistently higher rates of false positive SNPs compared to SKESA, inflating many final distance estimates beyond known true values. When working with assembly data alone, the robustness of any SNP distance estimation pipeline will rely on the accuracy of the underlying assembly. Application-specific validation of CSP2 for other data types or species would be best performed under the framework used here; combining simulated data and real-world data allows users to optimize the assembler choice, assembly parameters, and CSP2 QC metrics that result in output correlating best with a chosen benchmark method.

Wrapping CSP2 in Nextflow makes the addition of new analytical run modes and the integration of new data types straightforward. CSP2 does not currently include a tree-building step as part of SNP mode, but with Nextflow this is as simple as tacking on a module that calls a tree inference program such as IQ-TREE (*Minh et al., 2020*) or Randomized Axelerated Maximum Likelihood (RAxML) (*Stamatakis, 2014*) to the end of the current workflow. CSP1 maps reads using Bowtie2 (*Langdon, 2015*), but alternative mappers like *bbmap*, *minimap* (*Li, 2016*), or GPU-enabled options like GPU Accelerated Sequence Alignment Library v2 (GASAL2) (*Ahmed et al., 2019*) provide routes to re-incorporate read mapping back into CSP2 without significantly sacrificing speed. Read data could be used for direct estimation of SNP distances as in CSP1 or (*via* faster, high-stringency self-mapping) to pre-validate SNPs in assemblies prior to MUMmer alignment (*e.g.*, to confirm SNPs near contig edges, or to mask SNPs where minor allele frequencies fall above a threshold value). The presence or absence of plasmid data within pathogen assemblies can impact SNP distance estimates, especially when plasmids are present in most but not all assemblies (*Li et al., 2019*); incorporation of plasmid detection pipelines like plasmidSPAdes (*Antipov et al., 2016*) or plasmidFinder (*Carattoli & Hasman, 2020*) could both reduce error in distance estimation while also adding another dimension to CSP2 output through presence/absence data.

In this manuscript we leveraged the speed of CSP2 to analyze multiple reference genomes at once, ultimately choosing distance values based on the reference with the highest query alignment rates. While reasonable (and facilitating straightforward comparisons with single-reference pipelines like the NCBI Pathogen Detection pipeline or CSP1), a more nuanced approach to distance estimation is possible by incorporating data from multiple references into a single analysis. In addition to moving from point estimates towards distributions with confidence intervals, the integrated analysis of multiple reference genomes together would enable identification of reference-specific artifacts, and

also low-quality or error-rich queries that provide highly variable (or consistently aberrant) results across references.

## CONCLUSIONS

High-quality bacterial genome assemblies contain the required information to infer accurate SNP distances, and CSP2 provides a fast and versatile alternative for distance estimation. CSP2 distances were concordant with distances from CSP1 and the NCBI Pathogen Detection database. CSP2 provides many QC customization options to adapt for specific use-cases, and coding CSP2 in Nextflow allows for straightforward buildout of new functions.

## ACKNOWLEDGEMENTS

We thank A. Pightling for providing helpful feedback on drafts of this manuscript. We appreciate the support of M. Hammond and G. Engelbach for assistance with managing dependencies and the computational infrastructure on which the analyses were performed. Thanks to K. Kongati for discussion on configuring CSP2 to use Nextflow.

### Funding

The authors received no funding for this work.

### Competing Interests

The authors declare that they have no competing interests.

### Author Contributions

- Robert Literman conceived and designed the experiments, performed the experiments, analyzed the data, performed the computation work, prepared figures and/or tables, authored or reviewed drafts of the article, and approved the final draft.
- Jayanthi Gangiredla conceived and designed the experiments, authored or reviewed drafts of the article, and approved the final draft.
- Hugh Rand conceived and designed the experiments, authored or reviewed drafts of the article, and approved the final draft.
- James B. Pettengill conceived and designed the experiments, authored or reviewed drafts of the article, and approved the final draft.

### Data Availability

The code and raw data are available at GitHub and Zenodo:

- https://github.com/BobLiterman/CSP2_Manuscript_Code.
- Bob Literman. (2025). BobLiterman/CSP2_Manuscript_Code: Manuscript_Submission (Manuscript_Submission). Zenodo. https://doi.org/10.5281/zenodo.15076504.
- www.github.com/CFSAN-Biostatistics/CSP2

- Bob Literman, & Justin Payne. (2025). CFSAN-Biostatistics/CSP2: v.0.9.7.8 (v.0.9.7.8). Zenodo. https://doi.org/10.5281/zenodo.15741532.

## Supplemental Information

Supplemental information for this article can be found online at http://dx.doi.org/10.7717/peerj-cs.2878#supplemental-information.

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
