# Peer review of "CFSAN SNP Pipeline 2 (CSP2): a pipeline for fast and accurate SNP distance estimation from bacterial genome assemblies"

_PeerJ Computer Science, doi:10.7717/peerj-cs.2878_

## Round 0.1 · original submission · Major Revisions

· Academic Editor

Major Revisions

Dear authors,

You are advised to critically respond to all comments point by point when preparing an updated version of the manuscript and while preparing for the rebuttal letter. Please address all comments/suggestions provided by reviewers, considering that these should be added to the new version of the manuscript.

Kind regards,
PCoelho

·

Basic reporting

The authors use clear, unambiguous, and professional English throughout the manuscript.

The article's introduction for the most part provides sufficient context with relevant literature references to demonstrate how the work fits into the broader field of knowledge, but the overview of existing methods seems to only explicitly refer to the authors' own CSP1 pipeline (the predecessor to the CSP2 pipeline being introduced in this paper), whereas this is a fairly standard Bioinformatics problem (I work primarily in viral genomics, so not a direct one-to-one mapping, but there are *tons* of viral molecular epidemiology pipelines that do similar things), and I would be surprised if no other similar pipelines exist. Further, the introduction claims that "Mapping-based pipelines have been effective at uncovering the sources and modes of transmission for many pathogenic outbreaks [e.g., 4, 5], but the computational steps involved can be time-consuming and resource-intensive. The read files and the intermediate mapping files (e.g. SAM, BAM, and pileup files) can also result in a large storage footprint even when analyzing a small number of isolates", but I feel as though this sentence is misleading: (1) read mapping is *rarely* the computational bottleneck for such pipelines anymore with advancements in mapping tools like Minimap2 (which has presets for both short-read and long-read mapping), and (2) the pipeline the authors propose would not circumvent the issue of large storage footprint from storing raw FASTQ files (unless the authors throw away the FASTQs after assembly, which would be a terrible idea); on the contrary, after mapping, reads can be stored in a compressed BAM file or, better yet, a reference-based compressed CRAM file, to get significantly smaller storage footprint than would be required to store compressed FASTQ files. Further, you certainly don't need to store pile-up files (on the contrary, recent advances in tools like ViralConsensus can perform SNP calling by streaming reads directly on-the-fly, without any need for intermediate files like pile-ups). I would suggest the authors revise this section to be more representative of the actual bottlenecks of modern pathogen bioinformatics workflows.

The article has professional structure, figures, and tables. The tool itself is shared publicly on GitHub, but I'm unable to find the raw data and methods related to the benchmarking experiment performed in the paper: these data and any relevant analysis scripts/tools should be made available to ensure reproducibility.

The manuscript is self-contained and has relevant results to the hypotheses presented.

The formal results are clear, and all non-standard terms are defined.

Experimental design

The manuscript presents research that is within the Aims and Scope of the journal. However, I'm unsure of the originality of the research: CSP2 at a glance seems to essentially be the same pipeline as CSP1 (Davis et al, PeerJ Computer Science 2015; citation 3 in the paper), but with the only key difference being in how SNPs are called: CSP1 used Bowtie2 to map reads to the reference genome + Samtools to generate a pile-up from the mapped reads + Varscan to call SNPs from the pile-up, whereas CSP2 uses SPAdes or SKESA to perform de novo genome assembly + MUMmer to align the de novo assembly against a reference assembly and output the SNPs. After SNP calling, however, it's unclear how CSP1 and CSP2 differ: on the contrary, the authors say that, " After genome alignment, most data processing steps mirror the quality control measures used in the CFSAN SNP Pipeline (CSP1), including density filtering and handling missing data". Given that there are *many* different methods to call SNPs, I'm not sure swapping out just the SNP-calling procedure is sufficiently original. Since CSP2 uses Snakemake, and since the authors show differences in performance when using different SNP-calling pipelines, it would ideally be nice if the CSP2 pipeline could be flexible such that the user could choose from a wider range of SNP-calling approaches (e.g. VarScan, FreeBayes, iVar, ViralConsensus, etc.) and have the SKESA + MUMmer approach be the default.

The research question (whether this alternative "de novo assembly + full-genome alignment" SNP-calling approach will improve the overall pipeline performance) is well-defined, relevant, and meaningful. It is stated how this research fills the identified knowledge gap (via benchmarking experiment).

The investigation is performed to a high technical and ethical standard, though I think it could be more rigorous by evaluating other SNP-calling approaches.

The methods *within* the paper are described with sufficient detail, but given that the data + scripts/tools behind the benchmark experiment are not provided (at least I was unable to find them), the submission is not reproducible in its current state: those data + scripts/tools need to be published to make this work reproducible.

Validity of the findings

The findings are valid, though as I mentioned previously, it is unclear whether this is sufficiently different than CSP1. I am leaning towards "yes" because of the interesting results comparing SPAdes vs. SKESA to demonstrate how the choice of de novo assembler impacts accuracy, and because of the CSP1 vs. CSP2 vs. NCBI comparison.

The underlying data have *not* been provided: I can't seem to find the data + scripts/tools behind the benchmark experiment presented in the paper.

The conclusions are well-stated and are linked to the original research question, but they are *not* limited to supporting results. Specifically, the authors claim that "CSP2 generates accurate genetic distance estimates between bacterial pathogens using only genome assemblies, with faster runtimes and requiring fewer computational resources compared to traditional read mapping approaches", though the only read mapping approach they compare to is CSP1, which uses Bowtie2 (a relatively slow read mapper; there have been *significant* advances in the read mapping world in the past decade, e.g. Minimap2) and a pile-up approach to SNP-calling (a relatively slow and memory-intensive approach; small genomes like viruses and bacteria can be SNP-called on-the-fly using methods like ViralConsensus). In general, read mapping using modern tools (e.g. Minimap2) should be much faster than de novo assembly, so I would suggest removing any such claims about read mapping tools being too slow (unless the authors want to perform a full-fledged benchmark comparison against read-mapping approaches that utilize modern tools like Minimap2 and ViralConsensus).

For context, on my laptop (8-core w/ 8 GB RAM), I simulated reads from the NC_000913.3 E. coli reference genome at 30X coverage using ART:

art_illumina -ss HS25 -i NC_000913.3.fasta -l 150 -f 30 -o NC_000913.3.30X

And then performed mapping + SNP calling on-the-fly using Minimap2 v2.27-r1193 + ViralConsensus v0.0.6:

minimap2 -a -x sr NC_000913.3.fasta NC_000913.3.30X.fq | viral_consensus -i - -r NC_000913.3.fasta -o NC_000913.3.30X.consensus.fa -op NC_000913.3.30X.pos_counts.tsv -oi NC_000913.3.30X.ins_counts.json

The entire process took ~12 seconds, with Minimap2 reporting that read mapping took 9.5 seconds (though this is an *over*-estimate of how long read mapping actually took, as I was piping to ViralConsensus, so there is I/O blocking overhead). Importantly, this process scales linearly as a function of number of reads, so e.g. 300X coverage would likely take ~2 minutes total.

Reviewer 2 ·

Basic reporting

The manuscript describes the CFSAN SNP pipeline (CSP2), which extracts SNPs from genome assemblies using a reference-based approach. The resulting SNP calls can be stored in compact 'snpdiffs' files, which can be used to avoid re-analysis. The workflow has been tested on simulated and real data from several species and shows high agreement with read mapping-based approaches. The manuscript is well written and concise, and both the data and the code are publicly available. The figures are clear and well explained in the legend.

Experimental design

The manuscript describes the architecture of the workflow and some of the key parameters that can be tweaked. Afterwards, the validation on real world and simulated data are presented. The workflow is well documented and the interpretation is facilitated by Figure 1. The validation is clearly explained and the results are extensively documented in the main manuscript and the supplementary materials. The NCBI Pathogen Detection portal accession numbers for the validation data were clearly listed. Tool versions and parameters were all fully documented.

Validity of the findings

The authors demonstrate accurate SNP detection using CSP2 on datasets for four different species, using simulated and real world datasets. There were a relatively limited number of mismatches with read mapping based approaches, which were well-explained in the manuscript. In my opinion, the speedup over the mapping-based approach is very real, but perhaps a bit exaggerated in the manuscript. The de novo assembly and tree building itself can also be very time consuming (especially for larger datasets). In addition, could the authors clarify lines 333-334: “and the preservation of the raw alignment data in the snpdiffs file also reduces future runtimes as the alignment steps can be omitted completely.” Isn’t it the same for mapping-based approaches (i.e., storing VCF / BAM files)?

Additional comments

- Is there a way to mask known recombinant regions? I assume this is indirectly addressed by the 'ref_edge' parameter, as short-read assemblies typically break near recombinant repeat regions.
- I think the manuscript could benefit from a bit more explanation of the use of multiple reference genomes. I was not familiar with this concept and its benefits for the resulting SNP calls? Also, how are duplicate calls (i.e., SNP called in the same region in both reference genomes) handled?
- Can the authors provide some guidelines on how to construct a tree from the SNP calls reported by CSP2? Is there any reason to believe that these are different from the usual mapping-based approaches?
- Many labs are experimenting with long read Oxford Nanopore Technologies (ONT) sequencing for pathogen surveillance, especially with the recent increase in accuracy with the R10 sequencing chemistry. I think the manuscript could benefit from a small addition on why CSP2 is or is not suitable for the analysis of assemblies generated with ONT input data.
- How are contigs that originate from plasmids handled? Do the authors recommend a filtering step to remove these prior to running CSP2?
- Lines 369-372: Could the authors clarify this statement? If I understand correctly, the pipeline would use the AMR gene sequences as ‘reference genomes’? Would the tool work with many similar variants of the same gene. And does this refer to the detection of the genes themselves or only to the detection of specific point mutations within those genes?
- Line 137: Could you clarify the difference between missing and purged data here? I assume that purged data is positions that are removed due to the filter parameters?
- Line 175-176: I am not sure that a high contig count alone is sufficient as a quality check. I am aware that QC checks of assemblies are outside the scope of the manuscript, but perhaps the need for more extensive QC can be added to the discussion.
- Line 354-355: Not only the assembly pipeline, but also pre-processing steps such as DNA extraction, which could be a limiting factor in applying the workflow to datasets generated by different laboratories.

Reviewer 3 ·

Basic reporting

no comment

Experimental design

no comment

Validity of the findings

no comment

Additional comments

Literman et al. present an updated version of their CSP pipeline that is designed to work on assemblies, using MUMmer alignments to generate SNP distances. CSP2 is compared to both CSP1 and the NCBI pathogen detection pipeline using both simulated and real data. The software and documentation appears to be of high quality, and the comparisons indicate comparable accuracy across the three pipelines. Although CSP2 is not especially novel, it appears to be well designed and should be a useful addition to the pathogen bioinformatics toolkit.

Major comments:

My main concerns with the manuscript relate to the data analysis and presentation choices rather than the underlying pipeline design. For example, the figures showing the assembly metrics are quite thorough (Figs S1-4), but the same is not true for the pipeline accuracy analyses (Figs 2-3).

Line 115-116: –dwin and –wsnps - I am concerned that these defaults could remove a lot of real SNPs in regions of high SNP density. What was the rationale for choosing these, and have you tested the impact of different values?

Simulated data, line 214: The 350 bp cut off is responsible for most of the missed SNPs - what was the reasoning for using this as a cut off? It makes sense that this value becomes more of a factor with the more fragmented assemblies, as there are more edges. Did you try using different cut offs, and what was the impact on the number of missed SNPs vs false positive SNPs (presumably the reason for having this filter)?

Cluster concordance rates, line 181-182, 280: Why the choice of 3 SNPs to determine concordance? This is problematic as it does not take into account the genetic diversity within each cluster - 3 SNPs is a smaller error (per SNP) if two isolates are separated by 100 SNPs than if they are separated by 3 SNPs. In my opinion it would be much better to report the differences for each cluster (using boxplots, for example) as this may reveal cluster-specific patterns, and then summarise these across the species. Additionally, concordance could then be reported according to multiple thresholds, 1,2,3,4,5 etc, removing the dependence on this one value.

Figure 3B: I don’t think having a common y axis works for this plot. The scales are so different that there is essentially no resolution for the middle and right hand plots. There is also no y axis label. Separating these would enable both points to be addressed.

Minor comments:

Line 58: This is not necessarily true - Snippy, for example, will generate reads from an assembly and then proceed with a mapping based pipeline.

Line 64: “most” bacterial species - not all are haploid.

Line 66: It should be stated somewhere that generating the assemblies in the first place is much more computationally intensive than any mapping pipeline.

Line 262: State how many isolates for each species remained after the outlier filtering.

Line 279: Refer to A and B panels in figures, here and throughout.

Line 286: See point above about concordance rates - are the lower values between E.coli and the other species explained by relatively minor differences in the numbers of SNPs? For example, if concordance rates were ~90% using a threshold of 4 SNPs then this difference would not seem nearly as dramatic.

Line 295-298: These correlation coefficients and slopes would be much easier to interpret if scatter plots were presented. The values are high, but by themselves don’t reveal any patterns associated with certain clusters, or how this relates to cluster genetic diversity, for example.

Line 339: Again, the computational burden of generating the assemblies in the first place should be acknowledged.

Line 341: “versus hours” - what does this statement refer to?

Figure 3A: The title should make it clear that this is CSP1 vs CSP2.

---

## Round 0.2 · accepted · Accept

· Academic Editor

Accept

Dear authors, we are pleased to verify that you meet the reviewer's valuable feedback to improve your research.

Thank you for considering PeerJ Computer Science and submitting your work.

Kind regards
PCoelho

·

Basic reporting

The authors have addressed all of my previous concerns regarding Basic Reporting.

Experimental design

The authors have addressed all of my previous concerns regarding Experimental Design.

Validity of the findings

The authors have addressed all of my previous concerns regarding Validity of Findings.

Additional comments

The updated paper looks great!

Reviewer 2 ·

Basic reporting

No further comments.

Experimental design

No further comments.

Validity of the findings

No further comments.

Additional comments

No further comments.